# Prior cocaine use disrupts identification of hidden states by single units and neural ensembles in orbitofrontal cortex

**Wenhui Zong[1]\*, Lauren Mueller[1], Zhewei Zhang[1], Jinfeng Zhou[2]\*[†], Geoffrey Schoenbaum[1]\*[†]**

[1]National Institute on Drug Abuse, Intramural Research Program, Baltimore, United States; [2]State Key Laboratory of Cognitive Neuroscience and Learning, Beijing Normal University and Chinese Institute for Brain Research, Beijing, China

**\*For correspondence:**
wenhui.zong@nih.gov (WZ);
jingfengzhou@bnu.edu.cn (JZ);
geoffrey.schoenbaum@nih.gov
(GS)

[†]senior authors

**Competing interest:** The authors declare that no competing interests exist.

## eLife Assessment

This **fundamental** work shows that a history of cocaine self-administration disrupts the orbitofrontal cortex's ability to encode similarities between distinct sensory stimuli that possess identical task information—hidden states. The evidence supporting these conclusions is **compelling**, with methods and analyses spanning self-administration, a novel 'figure 8' sequential odor task, recordings from 3,881 single units, and sophisticated firing analyses revealing complex orbitofrontal representations of task structure. These results will be of broad interest to psychologists, neuroscientists, and clinicians.

**Abstract** The orbitofrontal cortex (OFC) is critical to identifying task structure and to generalizing appropriately across task states with similar underlying or hidden causes. This capability is at the heart of OFCs proposed role in a network responsible for cognitive mapping, and its loss can explain many deficits associated with OFC damage or inactivation. Substance use disorder is defined by behaviors that share much in common with these deficits, such as an inability to modify learned behaviors in the face of new information about undesired consequences. One explanation for this similarity would be if addictive drugs impacted the ability of OFC to recognize underlying similarities, hidden states, that allow information learned in one setting to be used in another. To explore this possibility, we trained rats to self-administer cocaine and then recorded single-unit activity in lateral OFC as these rats performed in an odor sequence task consisting of unique and shared positions. In well-trained controls, we observed chance decoding of sequence at shared positions and near chance decoding even at unique positions, reflecting the irrelevance of distinguishing these positions in the task. By contrast, in cocaine-experienced rats, decoding remained significantly elevated, particularly at the positions that had superficial sensory differences that were collapsed in controls across learning. These neural differences were accompanied by increases in behavioral variability at these positions. A tensor component analysis showed that this effect of reduced generalization after cocaine use also extended across positions in the sequences. These results show that prior cocaine use disrupts the normal identification of hidden states by OFC.

## Introduction

The orbitofrontal cortex (OFC) is essential for recognizing the underlying structure of tasks and for generalizing across contexts that share hidden or latent causes (*Moneta et al., 2024*; *Bein and Niv, 2025*; *Samborska et al., 2022*; *Farovik et al., 2015*; *Morrissey et al., 2017*; *Lin and Zhou, 2024*).

This capacity allows animals to infer the common features between seemingly different experiences and to adjust behavior accordingly—a process at the core of cognitive mapping (*Schuck et al., 2016*; *Wilson et al., 2014*). When this ability is disrupted by OFC lesions or inactivation, behavior can become overly tied to superficial idiosyncratic features, leading to inflexible or maladaptive responses when it is necessary to generalize across hidden states to update behavior (*Gardner and Schoenbaum, 2021*), as for example in behavioral settings such as after reversal or reinforcer devaluation or in sensory preconditioning (*Jentsch and Taylor, 1999*; *Panayi et al., 2024*; *Ersche et al., 2008*; *Ersche et al., 2016*; *Nelson and Killcross, 2006*; *Schoenbaum and Setlow, 2005*).

Notably, impairments in such OFC-dependent tasks have also been found to occur in experimental settings after experience with addictive drugs, particularly psychostimulants such as cocaine and amphetamine (*Ersche et al., 2016*; *Nelson and Killcross, 2006*; *Schoenbaum and Setlow, 2005*; *Jentsch et al., 2002*; *Groman et al., 2018*; *LeBlanc et al., 2013*; *LeBlanc et al., 2012*; *Simon et al., 2007*; *Calu et al., 2007*; *Wied et al., 2013*; *Lucantonio et al., 2014*; *Ersche et al., 2011*). Such results, coupled with evidence that addictive drugs affect markers of function in OFC and related areas (*Lucantonio et al., 2014*; *Stalnaker et al., 2007*; *Konova et al., 2012*; *Crombag et al., 2005*; *Wright et al., 2017*; *Mueller et al., 2021*; *Mueller et al., 2024*; *Takahashi et al., 2019*; *Parvaz et al., 2015*; *Konova et al., 2023*), suggest that drug-induced changes in OFC-dependent processing may underlie particularly pernicious features of substance use disorders, such as craving and relapse, in which maladaptive behaviors return despite treatment and even periods of abstinence (*Jentsch and Taylor, 1999*; *Lucantonio et al., 2012*; *Volkow and Fowler, 2000*). One possible explanation for this persistence is that chronic drug exposure compromises the OFC's ability to recognize hidden similarities between situations, thereby disrupting generalization across task states. Combined with pre-existing conditions and other environmental insults in certain individuals, such an effect could lead to the loss of behavioral control that characterizes addiction (*Pisupati et al., 2024*).

To investigate this possibility, we examined how prior cocaine use affects OFC representations of hidden states in a sequential decision-making task. Rats were trained to self-administer either cocaine or sucrose and were then recorded from the lateral OFC while performing an odor-based sequence task with positions that either shared or differed in sensory cues. This task design allowed us to assess whether OFC neurons appropriately compressed or generalized across positions with different sensory features but identical behavioral relevance—a hallmark of hidden state identification. As expected, OFC neurons in controls showed near-chance discrimination between comparable positions (P2 and P3) across sequences, reflecting compression of irrelevant sensory differences, and even at positions with unique sensory cues (P1 and P4), neural discrimination was near chance in most trial epochs. This neural compression reflects the OFC's preference to represent latent task states rather than external features. By contrast, cocaine-experienced rats maintained significantly higher selectivity at both shared and unique positions, failing to compress positions that were behaviorally equivalent. Their behavior also became more variable, suggesting reduced recognition of underlying equivalence across sequences. Tensor component analysis (TCA) showed that this loss of generalization extended beyond specific position pairs: cocaine-experienced rats lacked higher-order components that generalized across all positions, indicating a fundamental alteration in how OFC organizes task representations.

## Results

To examine the potential impact of cocaine use on the identification of hidden states by OFC neurons, we used a go, no-go odor discrimination task (*Figure 1A*). This task, used previously to record in OFC (*Zhou et al., 2021*), featured 6 odors arranged in two 4-odor sequences, labeled as S1 and S2 (*Figure 1B*). These sequences had unique odors at the start and end positions (P1 and P4) and shared odors at the two positions in the middle, where the sequences overlapped (P2 and P3). The experiment was conducted over a 4-month period consisting of several phases (*Figure 1C*). Initially, naive rats underwent a 4-week training on the figure 8 task. In the subsequent week, rats underwent jugular catheterization surgery and were given time to recover. For the following 2 weeks, rats were trained to self-administer either sucrose (10% wt/vol; *n* = 3) or cocaine (0.75 mg/kg/infusion; *n* = 3) (*Figure 1D*), using procedures similar to those known to induce incubation of craving (*Grimm et al., 2001*). The subsequent 3 weeks were dedicated to electrode implantation surgery in OFC and postsurgical recovery. Two additional weeks were then allocated for reminder training on the odor task.

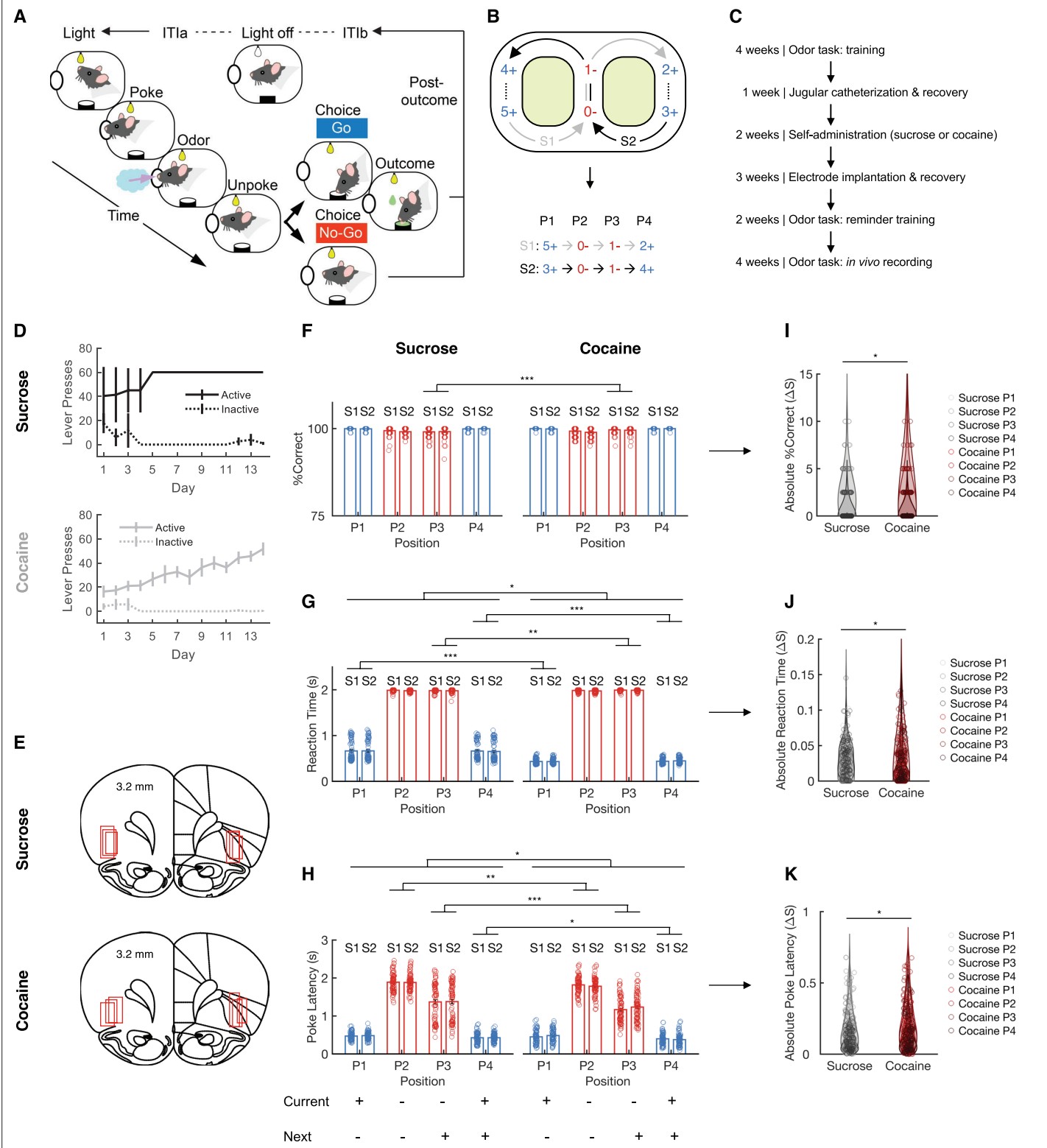

**Figure 1.** Figure 8 odor sequence task, cocaine self-administration, and behavior. (**A**) Schematic representation of the trial events in the odor sequence task. The initiation of each trial was indicated by the illumination of two overhead houselights. After poking into the central odor port and sampling the presented odor, rats had the option to respond with a 'go' to receive a sucrose reward or a 'no-go' to avoid a prolonged inter-trial interval. (**B**) The six odors were grouped into two sequences, S1 and S2, each consisting of four odor positions (P1–P4). The sequences alternated in a 'figure eight' pattern. The numbers at each position represent odor identities, and the blue/red symbols (+/−) indicate rewarded and non-rewarded trials. (**C**) Timeline of

*Figure 1 continued on next page*

*Figure 1 continued*

the experimental procedure. (**D**) The number of active (solid lines) and inactive (dashed lines) lever presses during sucrose self-administration sessions (upper; $n = 3$) and cocaine self-administration sessions (lower; $n = 3$). (**E**) Reconstruction of recording sites, with red squares indicating the locations of electrodes. (**F**) Percentage of correct responses (% correct) is shown for each trial type during single-unit recording sessions. A significant group difference emerged specifically at position P3 between the Sucrose ($n = 71$) and Cocaine ($n = 74$) groups ($F_{(1,288)} = 13.1$, $p = 3.4 \times 10^{-4}$, $\eta_p^2 = 0.043$; one-way ANOVA). Blue denotes rewarded trial types, while red denotes non-rewarded trial types. (**G**) Reaction time was defined as the interval between odor port exit and water well entry. For correct no-go trials, where no movement toward the water well was made, a fixed reaction time of 2 s—the full response window—was assigned. Reaction time differed significantly between the Sucrose ($n = 71$) and Cocaine ($n = 74$) groups overall ($F_{(1,1158)} = 6.6$, $p = 0.01$, $\eta_p^2 = 0.006$; one-way ANOVA), with position-specific analyses revealing significant differences at P1, P3, and P4 (P1: $F_{(1,288)} = 152.9$, $p = 1.9 \times 10^{-28}$, $\eta_p^2 = 0.35$; P3: $F_{(1,288)} = 9.3$, $p = 2.5 \times 10^{-3}$, $\eta_p^2 = 0.03$; P4: $F_{(1,288)} = 139.6$, $p = 1.6 \times 10^{-26}$, $\eta_p^2 = 0.33$; one-way ANOVA). (**H**) Poke latency, measured as the time from light onset to odor port entry. Significant group differences were found across positions ($F_{(1,1158)} = 4.0$, $p = 0.04$, $\eta_p^2 = 0.004$; one-way ANOVA), with specific differences emerging at positions P2, P3, and P4 (P2: $F_{(1,288)} = 10.0$, $p = 1.7 \times 10^{-3}$, $\eta_p^2 = 0.034$; P3: $F_{(1,288)} = 12.4$, $p = 5.0 \times 10^{-4}$, $\eta_p^2 = 0.04$; P4: $F_{(1,288)} = 5.7$, $p = 0.018$, $\eta_p^2 = 0.02$; one-way ANOVA, $n = 71$ (Sucrose) and $n = 74$ (Cocaine)). (**I**) A two-way ANOVA revealed significant main effects of group ($F_{(1,572)} = 4.0$, $p = 0.045$, $\eta_p^2 = 7.0 \times 10^{-3}$) on the absolute difference in percent correct between S1 and S2 for the Sucrose ($n = 71$) and Cocaine ($n = 74$) groups. Sucrose data are shown as black color with increasing shading intensity from P1 to P4, while Cocaine data are shown as red color with similarly graded shading from P1 to P4. (**J**) For the absolute S2–S1 difference in reaction time across positions between Sucrose ($n = 71$) and Cocaine ($n = 74$) groups, two-way ANOVA revealed significant main effects of group ($F_{(1,572)} = 6.4$, $p = 0.012$, $\eta_p^2 = 0.01$) and position ($F_{(3,572)} = 3.3$, $p = 0.021$, $\eta_p^2 = 0.017$). Black circles denote sucrose data, with shading that deepens progressively from P1 through P4. Cocaine data are indicated by red circles, with coloration shifting from light to dark red across positions P1–P4. (**K**) Two-way ANOVA revealed significant main effects of group ($F_{(1,572)} = 5.6$, $p = 0.018$, $\eta_p^2 = 0.01$) and position ($F_{(3,572)} = 37.7$, $p = 3.2 \times 10^{-22}$, $\eta_p^2 = 0.16$) on the absolute poke latency difference between S1 and S2 in Sucrose ($n = 71$) and Cocaine ($n = 74$) groups. Sucrose data are plotted as graded black circles, with shading progressing from light to dark across positions P1–P4. Cocaine data are plotted as graded red circles, with shading progressing from light red to dark red across positions P1–P4. Error bars represent standard errors of the mean (SEMs). *$p < 0.05$; **$p < 0.01$; ***$p < 0.001$.

Finally, in vivo recordings were obtained during the last 4 weeks while the rats performed the odor task.

During these recording sessions, rats in both groups displayed consistently high levels of discrimination performance across all positions in the odor sequence (*Figure 1F*). In this study, rats were first trained on the odor sequence task to criterion before undergoing cocaine self-administration. Thus, all animals had already acquired the task thoroughly prior to drug exposure, and the task itself was relatively simple. Nevertheless, we observed several robust and significant differences between the cocaine- and sucrose-trained groups. Differences in trial initiation latencies were observed (*Figure 1H*), suggesting that the rats in both groups utilized the predictable odor sequence to anticipate the availability of reward not only in the current trial but also in subsequent trials. ANOVAs revealed a variety of small but significant group differences in these measures (*Figure 1F–H*). In addition to these somewhat idiosyncratic differences, analyses also showed effects in line with the proposal that past cocaine use had affected the rats' ability to ignore external differences between positions across the two sequences. Specifically, the behavior of the rats that had self-administered cocaine was more variable across sequences. This was evident in the distribution of difference scores between positions in each sequence on all three measures, which were larger in the cocaine group than in controls (*Figure 1I–K*). Thus, even after extensive training, animals with a history of cocaine exposure maintained a stronger behavioral distinction between the same position in the two sequences, in all three behavioral measures.

## Prior cocaine use disrupts identification of hidden states across position pairs

During these sessions, we recorded a total of 1699 units in the lateral OFC of three rats in the cocaine group and 2182 units from three rats in the control group (*Figure 1E*). A cursory examination of single-cell examples (*Figure 2A–D*) indicated that prior cocaine use was associated with less uniform neural activity at individual positions across sequences. To investigate this effect, we tested whether cocaine use affected the ability of single units and ensembles in the OFC to discriminate between positions in the task, starting with comparisons between the same positions on sequences S1 and S2. Neuronal selectivity on trials at comparable positions in the two sequences was calculated for each trial epoch, and the results showed that the proportion of selective units was very low—essentially at chance during trials at P2 and P3—before increasing at the end of P3 and into the odor sampling periods of P4 and P1. This pattern reflected compression of positions in the central arm

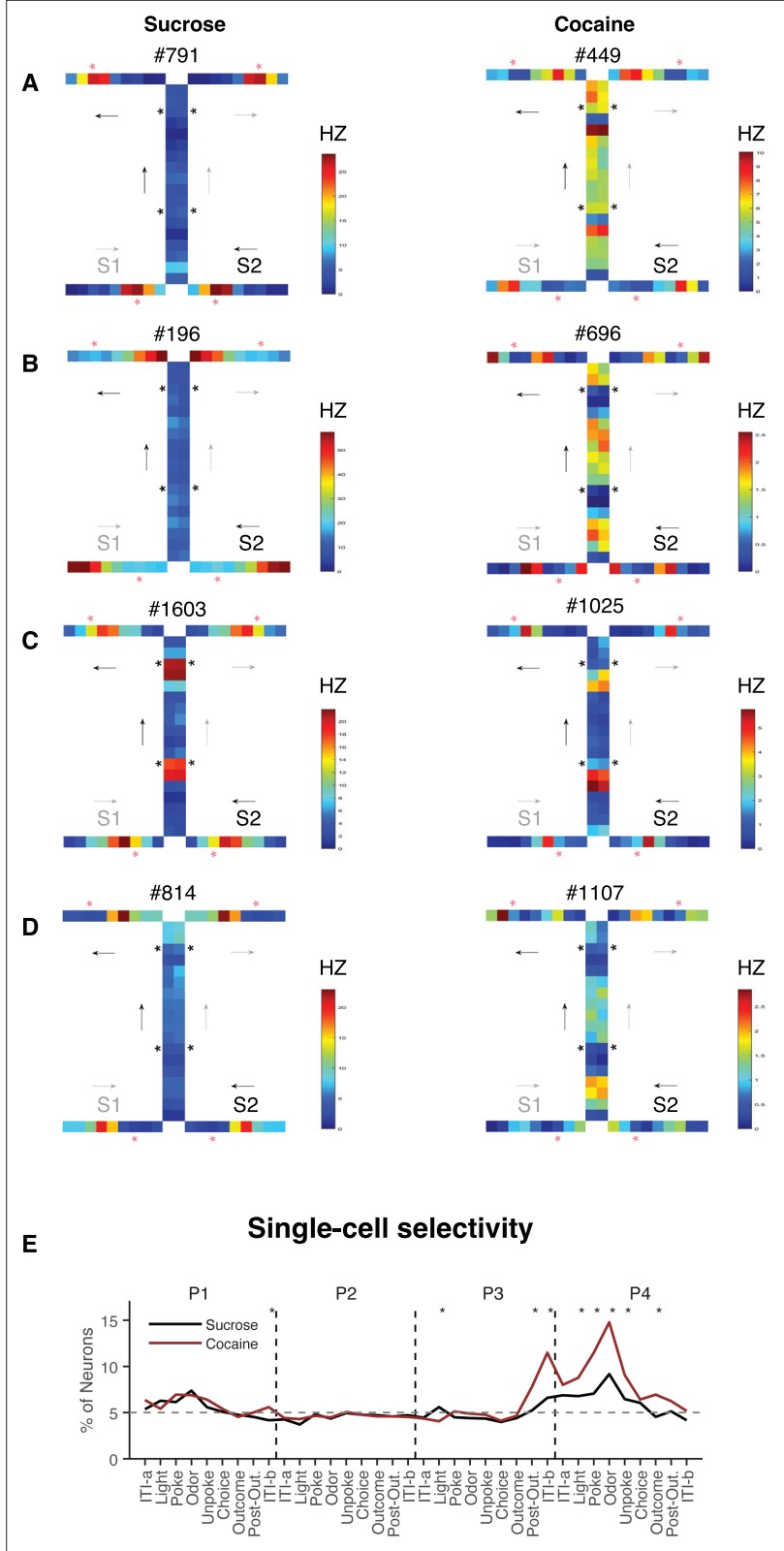

**Figure 2.** Discriminating sequences S1 versus S2 at single-unit level. (**A–D**) Heatmaps showing the activity of individual orbitofrontal cortex (OFC) neurons. Each panel illustrates the firing pattern of a single neuron, capturing its response dynamics across task epochs. Each small grid represents one epoch at a specific position, arranged in the direction of the arrow following the sequence: ['ITI-a', 'Light', 'Poke', 'Odor', 'Unpoke', 'Choice',

*Figure 2 continued on next page*

*Figure 2 continued*

'Outcome', 'Post-Out', 'ITI-b'], repeated for all four positions of each sequence. The left columns represent single OFC neurons from the Sucrose group, where the intensity of left-side epochs closely matches that of the right-side epochs, indicating strong generalization. In contrast, the Cocaine group shows comparatively weaker generalization across sides. S1, sequence 1; S2, sequence 2. (**E**) Line plot showing neuronal selectivity at each position in S1 versus S2 for each neuron in both the Sucrose ($n$ = 2182, black line) and Cocaine ($n$ = 1699, red line) groups. The selectivity was calculated at different task epochs for all four positions. Each asterisk indicates that there are significant differences between the two groups ($\chi^2$'s > 4.8; p's < 0.03; Chi-squared test). *p < 0.05; error bars are SEMs.

of the figure-8 maze, where the two sequences were identical, and discrimination of positions on the outer arms of the maze, where the odors differed. Notably, however, even for P4 and P1, most epochs within the trial showed near chance levels of selectivity in controls, while neurons recorded in cocaine rats maintained significantly higher levels of selectivity (compared both to controls and chance) at the end of P3 and during most of the epochs in P4 when compared directly between the two groups (*Figure 2E*).

To quantify this effect, we conducted a two-way ANOVA on the activity of each neuron at each position in the two sequences, with sequence and position as factors. This analysis revealed disproportionate effects of cocaine on the prevalence of neurons showing a significant interaction between these two factors (*Figure 3—figure supplement 1*), suggesting increased divergence in neural activity at similar positions across sequences in the cocaine group. This impression was reinforced by a second analysis, in which we correlated the preferred position of each neuron in each epoch across sequences. For this, we first identified position-selective neurons independently at each epoch and on each sequence (ANOVA, p < 0.01). A neuron's preferred position was then taken as the position with the highest firing, and then we calculated a correlation coefficient for each group across all neurons and epochs. As shown in the plot, the Sucrose group exhibited a steeper correlation compared to the Cocaine group. To statistically compare the correlation coefficients, we used Fisher's *r*-to-*z* transformation and found a significant difference between the groups (*Figure 3—figure supplement 2*).

The increase in differential activity across sequences in the cocaine group was also evident in an ensemble decoding analysis (*Figure 3A*). Decoding accuracy was above chance for most epochs at positions P1 and P4 and at the end of the trial and into the ITI period at P3, and while this was true in both groups, the decoding accuracy was significantly higher in the cocaine group in nearly all epochs. This difference between groups was even more evident when average decoding across epochs at each position was directly compared between the two groups (*Figure 3B*). Thus, both single unit and ensemble activity in lateral OFC in cocaine-experienced rats compressed meaningful task epochs less than in sucrose-trained controls. Additional analyses examining decoding across all positions within- and across-sequences showed that the Cocaine group exhibited significantly higher decoding accuracy within-sequence and significantly lower decoding accuracy across-sequence, compared to the Sucrose group (*Figure 3D–G*).

We propose that OFC representations develop through multiple stages, progressing from sensory- and reinforcement-driven coding to sequence-dependent differentiation as animals acquire task structure. With extended training, these representations are normally refined such that functionally equivalent states are compressed, and behaviorally irrelevant distinctions are suppressed. Prior cocaine exposure appears to disrupt this later refinement stage, leaving OFC representations in an earlier, sequence-specific state despite extensive training, consistent with impaired generalization across latent task states. Accordingly, the effect of cocaine was similar to the effect of diminished training on the task. This is evident in a comparison to data from OFC in rats in a prior study using the same task (*Zhou et al., 2021*), in which there was much less training prior to recording (6 weeks of odor task training in total plus 2 weeks of self-administration in this study vs. ~3 weeks of odor task training in the prior study). OFC neurons recorded in these rats revealed preserved decoding relative to the over-trained controls in the current study, decoding that was similar to that in the cocaine rats (*Figure 3C*). Thus, neural activity in OFC evolves during learning to identify the underlying hidden states that define behavioral relevance. This refinement or its specificity is disrupted by prior cocaine use.

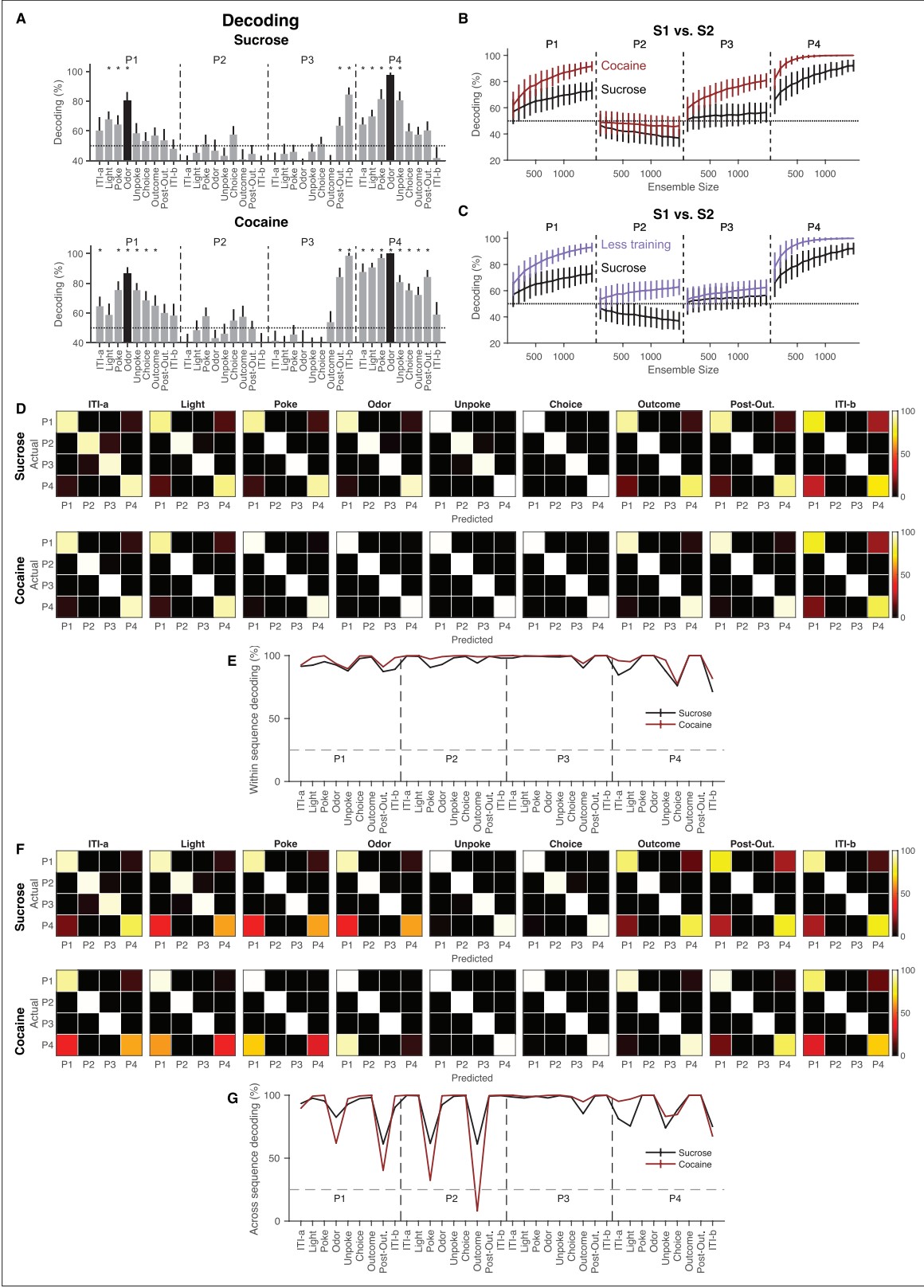

**Figure 3.** Cocaine use reduces the ability of orbitofrontal cortex (OFC) to generalize task-irrelevant odor-overlapping sequences. (**A**) Decoding accuracy of S1 versus S2 was evaluated within each of the nine task epochs for positions P1–P4. Error bars represent SDs, and each asterisk indicates that the mean decoding accuracy exceeds a 95% confidence interval calculated using the same decoding process with label-shuffled data. The dotted lines represent the chance level of decoding. The meaning of the black bars is consistent with *Figure 2E* (n = 2182 (Sucrose) and n = 1699 (Cocaine)).

*Figure 3 continued on next page*

*Figure 3 continued*

(**B**) Decoding accuracy of S1 versus S2 at each position was assessed using varying ensemble sizes for the Sucrose (black) and Cocaine (red) groups across all epochs. The Cocaine group demonstrated higher decoding accuracy and reduced generalization of task-irrelevant, overlapping sequences compared to the Sucrose group ($F_{(1,118)} = 16.8$, $p = 7.8 \times 10^{-5}$, $\eta_p^2 = 0.12$; one-way ANOVA, n = 1000 (Sucrose) and n = 1000 (Cocaine)). P1, P2, P3, and P4 represent positions 1, 2, 3, and 4, respectively. (**C**) Decoding accuracy of S1 versus S2 at each position was assessed using varying ensemble sizes for the Sucrose group (n = 1000, black, same data as in **B**) and the Less Training group (n = 1000, blue, *Zhou et al., 2021*) across all epochs. The decoding accuracy in the Less Training group was comparable to that observed in the Cocaine group (**B**, red) ($F_{(1,118)} = 0.01$, $p = 0.92$, $\eta_p^2 = 9.0 \times 10^{-5}$; one-way ANOVA). Additionally, the Less Training group exhibited significantly higher decoding accuracy than the Sucrose group ($F_{(1,118)} = 18.1$, $p = 4.2 \times 10^{-5}$, $\eta_p^2 = 0.13$; one-way ANOVA). P1, P2, P3, and P4 represent positions 1, 2, 3, and 4, respectively. *p < 0.05; error bars indicate SDs. (**D**) Confusion matrices showing within-sequence position decoding from OFC ensemble activity at four task positions for the Sucrose and Cocaine groups. The *y*-axis denotes the rats' actual position, and the *x*-axis indicates the predicted position. Brighter colors reflect higher decoding probabilities. (**E**) Quantification of within-sequence decoding accuracy across positions revealed a significant difference between groups (p = 0.027; W = 1118; two-sided Wilcoxon rank-sum test), with Sucrose (n = 1000) shown in black and Cocaine (n = 1000) in red. (**F**) Confusion matrices showing across-sequence position decoding from OFC ensemble activity at the same four task positions. Axes are as in (**D**), with brighter colors indicating higher decoding probabilities. (**G**) Quantification of across-sequence decoding accuracy also revealed a significant group difference (p = 0.046; W = 1144; two-sided Wilcoxon rank-sum test), with Sucrose (n = 1000) shown in black and Cocaine (n = 1000) in red.

The online version of this article includes the following figure supplement(s) for figure 3:

**Figure supplement 1.** Two-way ANOVA of sequence and position effects.

**Figure supplement 2.** Correlation of preferred positions.

## Prior cocaine use disrupts identification of hidden states across all positions

While the planned analyses conducted above show that prior cocaine use was associated with preserved discrimination of incidental information about task positions normally compressed by rats performing this task, they do not address whether this is a general effect or whether it only impacted direct comparison of the position pairs highlighted by the analyses. That is, did neurons in the cocaine-experienced rats maintain information about these positions because of the idiosyncratic features of our task, or is there a general effect of cocaine on the ability of the OFC to register common features and underlying causes. To get at this more general question, we utilized TCA to identify the dominant factors shaping neural activity across units, epochs, and trials in our task. This analysis provides a less constrained, more hypothesis-agnostic approach to our question since it looks for patterns across all of these factors in the design. If cocaine is causing a failure of generalization more broadly, then we would expect loss of power in dimensions across positions and not just within position pairs.

The results for this analysis are shown in *Figure 4A, B* and *Figure 4—figure supplement 1*. Neural variance was partitioned into components, one per row. The analysis shown was constrained to 10 components, as this number produced the most consistent and reliable results compared to other choices (*Figure 4—figure supplements 2 and 3*); additionally, the features we will highlight for each group were generally consistent within individual rats from each group (*Figure 4—figure supplements 4 and 5*). For each group, the characteristics of each component are visualized by the contributing neuron weight (left), temporal dynamics within a trial (middle), and amplitude of such dynamics across trials organized by positions (right). The temporal factor captures the activity pattern across epochs within a trial, whereas the trial factor reflects how strongly that pattern is expressed across positions and trials. Because TCA components are scale-indeterminate, factor magnitudes are meaningful only relative to one another within a component, not across components. Thus, differences in trial factors with similar temporal dynamics indicate differential recruitment of the same within-trial activity pattern across task positions, rather than changes in response timing. To discuss one example, the top row in the control group shows that the first component was broadly distributed across neurons (left panel), expressed most strongly at epochs of Unpoke, Choice, and Outcome (middle panel), and exhibited high amplitude at P1 and P4 and low at P2 and P3 (right panel). Further, it was higher in P3 than P2 and also showed similar patterns across trials within each of these position pairs (right panel).

After aligning these component patterns between groups based on their temporal factors, we observed different patterns in trial amplitudes reflecting differences in positional representations (right columns). In controls, most of the trial factors distinguished between the position pairs (rows 1–3, 6–8, and 10). These factors distinguished rewarded from non-rewarded position pairs, in some cases further distinguishing particular position pairs within each category (rows 2, 3, 7, and 8) and/or

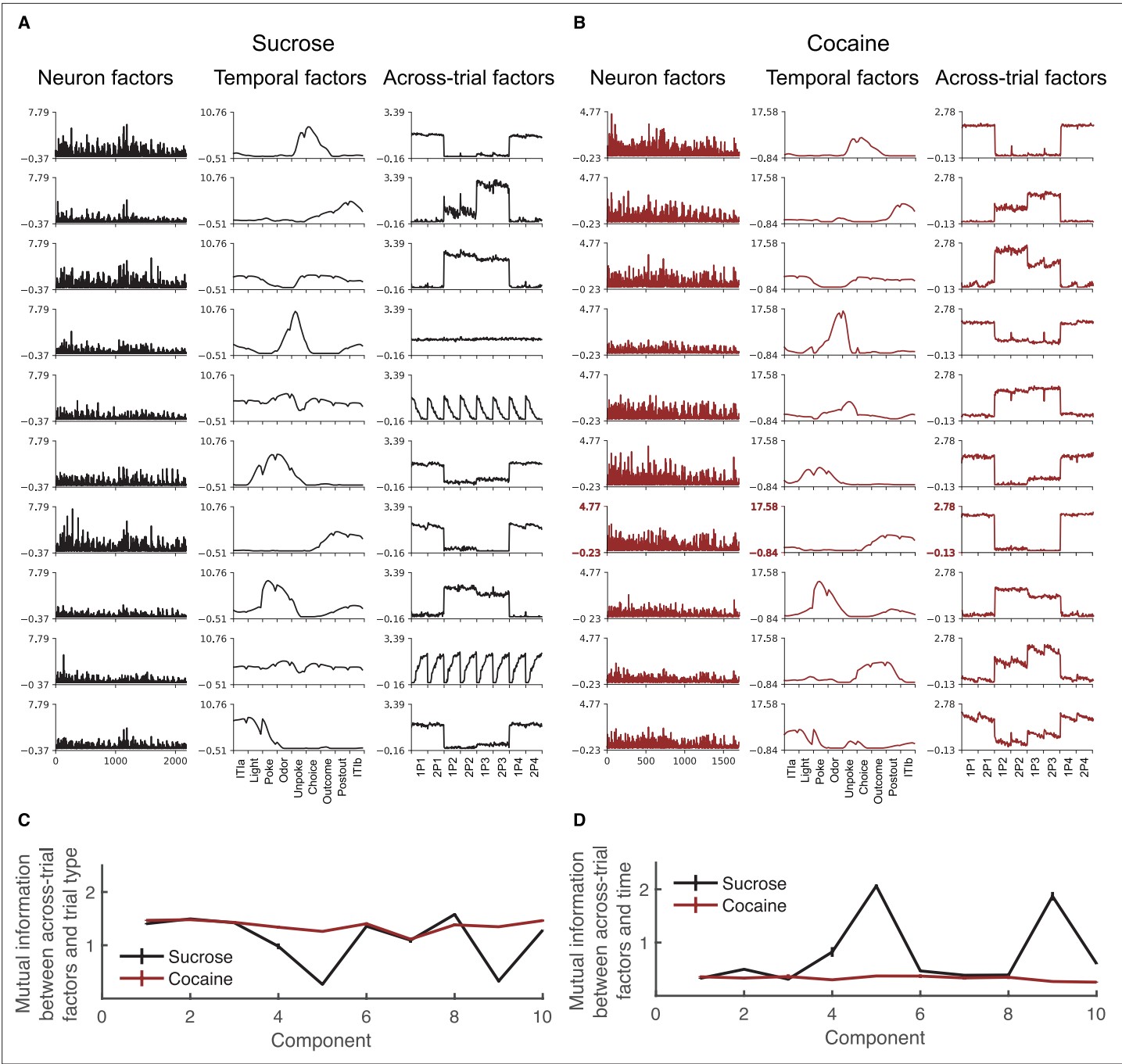

**Figure 4.** Cocaine use impairs the OFC's function in temporal cognition and decreases the generalization of task-irrelevant odor-overlapping sequences. Ten components of tensor component analysis (TCA) applied to sucrose (**A**) and cocaine (**B**) neuron activity are displayed. Each component consists of a neuron factor (left column), a temporal factor spanning nine events with eight time points per event (middle column), and a trial factor (right column). Temporal factors are grouped as follows: 1–8 (preTrial), 9–16 (Light), 17–24 (Poke), 25–32 (Odor), 33–40 (Unpoke), 41–48 (Choice), 49–56 (Outcome), 57–64 (postTrial1), and 65–72 (postTrial2). The dataset includes eight trial types, each with 40 trials. Trial factors 1–80 and 241–320 represent rewarded (positive) trials, while 81–240 correspond to unrewarded (negative) trials. Trials are further categorized as follows: 1–80 (P1), 81–160 (P2), 161–240 (P3), and 241–320 (P4). These low-dimensional components were extracted from a 10-component model for both the Sucrose (**A**) and Cocaine (**B**) groups. In the Sucrose group, components 1, 6, 7, and 10 encode positive trials, while components 2, 3, and 8 encode negative trials. Similarly, in the Cocaine group, components 1, 4, 6, 7, and 10 encode positive trials, whereas components 2, 3, 5, 8, and 9 encode negative trials. Notably, in the Sucrose group, beyond reward-related components, additional components encode aspects of temporal cognition (components 5 and 9), along with early components broadly active across most trials (component 4). (**C**) Plot illustrating mutual information (MI) between across-trial factors and trial types. The Sucrose group exhibits lower MI, particularly at components 5 and 9. A two-way ANOVA confirmed a significant reduction in MI for the Sucrose

*Figure 4 continued on next page*

*Figure 4 continued*

group ($F_{(1,1360)}$ = 571.6, p = 9.7 × 10$^{-106}$, $\eta_p^2$ = 0.3, n = 100 (Sucrose) and n = 100 (Cocaine)), especially at these components, which showed consistent patterns across all trial types, aligning with the plots in (**A**) of the Sucrose group, indicating a weaker dependency between across-trial factors and trial types. A significant difference was also observed across components ($F_{(9,1360)}$ = 266.3, p = 1.2 × 10$^{-292}$, $\eta_p^2$ = 0.64, n = 10). (**D**) Plot illustrating MI between across-trial factors and time. The Sucrose group demonstrates greater MI, particularly at components 5 and 9, suggesting a stronger dependency between across-trial factors and time. A two-way ANOVA confirmed a significant increase in MI for the Sucrose group ($F_{(1,1360)}$ = 850.1, p = 1.4 × 10$^{-145}$, $\eta_p^2$ = 0.38, n = 100 (Sucrose) and n = 100 (Cocaine)). Additionally, a significant difference was observed across components ($F_{(9,1360)}$ = 176.8, p = 1.4 × 10$^{-221}$, $\eta_p^2$ = 0.54, n = 10). Error bars are SEM.

The online version of this article includes the following figure supplement(s) for figure 4:

**Figure supplement 1.** Error and similarity plots for tensor component analysis (TCA).

**Figure supplement 2.** Four-component model, cocaine impairs orbitofrontal cortex (OFC) temporal cognition and sequence generalization with irrelevant overlapping odors.

**Figure supplement 3.** Fourteen component model, cocaine use impairs orbitofrontal cortex (OFC) function in temporal cognition and hinders generalization across odor-overlapping, task-irrelevant sequences.

**Figure supplement 4.** Ten-component model for individual Sucrose rats.

**Figure supplement 5.** Ten-component tensor component analysis (TCA) model applied to individual rats from the Cocaine group.

showing effects of trials (rows 2 and 3). However, none showed differences between positions or trials inside a pair. This result is consistent with findings in *Figures 2 and 3*, which showed that both individual units and ensemble responses in control rats collapsed the positions within each pair. However, controls also had three additional factors that appeared to generalize not just across positions inside a pair but rather across all positions (rows 4, 5, and 9). These factors were identical across all 8 positions and two (rows 5 and 9) showed identical changes across trials within each position. This indicates a significant tendency in neural activity in OFC to represent commonalities across all eight positions dynamically across time in a session; this global generalization goes beyond that illustrated in our planned comparisons in *Figures 2 and 3*.

The trial factor patterns in the cocaine-experienced rats were substantially different; all of the 10 factors distinguished rewarded versus non-rewarded position pairs, and the position-general factors evident in controls (rows 4, 5, and 9) were absent from activity in the cocaine rats. This dichotomy is consistent with reduced compression evident in our planned comparisons in *Figures 2 and 3* and further points to diminished generalization across task features and a heightened emphasis on differences across positions in the task, which presumably reflects the different values imparted by the sequences. These general features were robust and interpretable across animals in both the Sucrose and cocaine-exposed groups (*Figure 4—figure supplements 4 and 5*). This consistency supports the validity of comparing TCA-derived measures across groups, and consistency across individuals within each group.

To quantify the differences in *Figure 4A, B*, we used mutual information (MI) to measure the information available in each factor—particularly factors 5 and 9 about position and trial. Consistent with the description above, we found lower MI values for position in controls at these components compared to the cocaine group (*Figure 4C*) and higher MI at the same components for trial or temporal information (*Figure 4D*). Moreover, similar results were observed across varying numbers of components, with the significant difference between the two groups remaining consistent regardless of the number of components selected (*Figure 4—figure supplements 2 and 3*).

## Discussion

Prior to considering the implications of the current results, several limitations should be noted. First, sucrose self-administration is not a neutral manipulation and may itself promote abstraction in OFC representations by providing additional instrumental experience with the same reinforcer used in the odor-guided task. Second, the number of animals per group was relatively small, and all subjects were male, limiting the ability to fully assess individual variability, potential sex-dependent effects, or differences in motivational state or attentional engagement that may have contributed to the observed effects. Nonetheless, the key neural and behavioral signatures were consistent across individuals and analytic approaches, with no outliers observed, and sample sizes of this scale are common in cocaine self-administration studies due to their technical and logistical demands.

These results are consistent with a cocaine-induced disruption of the normal identification of underlying hidden states by OFC (*Moneta et al., 2024*; *Bein and Niv, 2025*; *Samborska et al., 2022*; *Farovik et al., 2015*; *Morrissey et al., 2017*; *Lin and Zhou, 2024*). Specifically, in the current task, OFC representations normally evolve with training to compress or generalize similar positions in the two sequences, even when external sensory information—the odor cues—differ. Cocaine-experienced rats failed to show this normal compression of irrelevant information, instead discriminating these position pairs at higher rates, similar to encoding observed in rats early in learning. Additionally, this failure to generalize was also evident in a TCA analysis, where factors generalizing across different positions were prominent in control data but entirely absent in data from cocaine-experienced rats.

The loss of this normal function of OFC is relevant to addiction, since it suggests that some addictive drugs, at least the psychostimulants, cause fundamental and long-lasting changes in how prefrontal areas process task-related information. These effects are consistent with prior studies showing changes in OFC function after drug use, particularly for behaviors in which it is necessary to generalize across hidden task states to appropriately update behavior when likely outcomes change, such as after devaluation and in sensory preconditioning or even reversal (*Jentsch and Taylor, 1999*; *Panayi et al., 2024*; *Ersche et al., 2008*; *Ersche et al., 2016*; *Nelson and Killcross, 2006*; *Schoenbaum and Setlow, 2005*). In rodents, drug-induced changes in OFC-dependent learning and insight are accompanied by long-lasting alterations in OFC neural activity, including degraded single-unit encoding of expected outcomes, altered ensemble representations of task structure, and persistent changes in other properties of OFC neurons (*Schoenbaum and Setlow, 2005*; *Wied et al., 2013*; *Lucantonio et al., 2014*; *Crombag et al., 2005*; *Wright et al., 2017*; *Mueller et al., 2024*). Convergent findings in nonhuman primates and humans further demonstrate that drug exposure is associated with OFC dysfunction and inflexible choice behavior (*Ersche et al., 2016*; *Jentsch et al., 2002*; *Ersche et al., 2011*; *Goldstein et al., 2001*), supporting a conserved role for OFC across species in guiding adaptive decision-making. Together, this body of work also suggests that cocaine self-administration induces enduring changes in OFC computations that undermine generalization across hidden task states, providing a mechanistic link between altered neural coding in OFC and the behavioral inflexibility that characterizes addiction.

Consistent with such speculation, the current findings show a specific effect of cocaine use on generalization or the ability of OFC neurons to collapse trivial external information to encode underlying causes that different situations have in common (*Pisupati et al., 2024*). A loss of this ability could have wide-ranging effects (*Radulescu and Niv, 2019*), but it might particularly disrupt the mobilization of learning from other settings to counteract or diminish drug-seeking behaviors. For instance, consequences of drug use learned in separate contexts or situations—for instance at home or in the classroom, during counseling, or even from observing the impact of drug use in the lives of others—would not be as effectively deployed to affect behavior during one's own drug-seeking. Similarly, therapeutic approaches designed to extinguish drug-seeking would also generalize more poorly outside of the clinical setting. The present results identify a neurophysiological mechanism—loss of OFC-mediated generalization—that may underlie the persistence and context-specificity of drug-seeking behavior, providing a new window into how addictive drugs alter cognitive mapping and flexible decision-making in the brain.

## Methods

### Contact for reagent and resource sharing

This study did not generate any unique reagents. However, for further information or requests for resources and reagents, please contact the Lead Contact, Geoffrey Schoenbaum (geoffrey.schoenbaum@nih.gov).

### Experimental model and subject details

The study was conducted on a group of six male Long-Evans rats (Charles River strain), aged around 3 months and weighing between 175 and 200 g. The rats were housed individually in an AAALAC-accredited animal care facility at the National Institute on Drug Abuse Intramural Research Program (NIDA-IRP), with ad libitum access to food on a 12-hr light–dark cycle. Water was removed a day before testing, and the rats were provided with free access to water for 10 min each afternoon in their

home cages. If there was no testing scheduled for the following day, they were given free access to water. All behavioral testing was conducted at the NIDA-IRP, and the animal care and experimental procedures were conducted in accordance with the guidelines set by the US National Institutes of Health (NIH) and approved by the Animal Care and Use Committee (ACUC) at the NIDA-IRP under Protocol No. 19-CNRB-108.

## Method details

### Figure 8 task

The study employed aluminum boxes (18 in. on a side) equipped with a port for odor delivery and a well for delivery of sucrose solution for conducting behavioral training. A custom-written C++ program and a system of relays and solenoid valves were used to control the task events. The entries into the odor port and the fluid well were detected by infrared beam sensors. The availability of each trial was signaled by the illumination of two house lights above the odor port. The trial was initiated if the rat entered the odor port within 5 s after light onset, leading to odor delivery after a 500-ms delay. The rats were required to remain in the port for an additional 500 ms; otherwise, the trial was aborted, and the lights extinguished. After 500 ms, the rats were free to leave the port, terminating odor delivery. Post port exit, the rats had 2 s to respond at the fluid well. Responding on rewarded trials led to the delivery of a sucrose solution (10% wt/vol; 50 µl) after a random delay ranging from 400 to 1500 ms. On non-rewarded trials, nonresponding during the 2-s period, or responding after the 2 s, the house lights were extinguished, indicating the end of the trial and the beginning of the ITI. Correct trials were followed by a 4-s ITI, and trials on which the rat made an error were followed by an 8-s ITI.

The study for the Figure 8 task included six odors, organized into two sequences (S1 and S2) that occurred repeatedly in turn (S1 → S2 → S1 → S2 → … → S1 → S2; 40 repeats of each sequence). On each trial, one of six odors was delivered to the odor port. The odor identity is indicated by a number, and reward and non-rewarded are indicated by the positive (+) and negative (–) symbols, orders of the odors were organized and shown below:

> S1: 5+, 0–, 1–, 2+
> S2: 3+, 0–, 1–, 4+

To avoid bias, the starting sequence for each session—either S1 or S2—was determined in a fully pseudorandom manner. Before training with any odors, rats were first shaped to nosepoke at the odor port and then respond at the well for a reward. The rats were trained on the full set of sequences from Day 1 until they achieved >75% accuracy on every trial type in a session. Following this, electrode arrays were implanted bilaterally in the OFC.

To clarify, the animals were not initially trained on the Figure 8 task (which involves six odors across two four-odor sequences, S1 and S2) prior to exposure to the full 24-position task. Each rat in this study was trained and recorded on only one behavioral paradigm—either the Figure 8 task or the 24-Position Odor Sequence Task, but not both. These tasks were conducted in separate cohorts of animals. All data presented in the figures of this manuscript were obtained from the Figure 8 task, using sucrose-trained, cocaine-trained, or previously reported minimally trained animals, with the exception of *Figure 3—figure supplement 1*. The data shown in *Figure 3—figure supplement 1*, derived from the 24-Position Odor Sequence Task, were published previously, and the behavioral paradigm is described in the corresponding figure legend.

### Surgical procedures

Rats were surgically implanted with a total of 32 electrodes, organized into two bundles of 16 electrodes each. These bundles were constructed using nickel–chromium wires with a bare diameter of 25 µm (AM Systems, WA). The implantation targeted the bilateral orbitofrontal cortices (AP: 3 mm, ML: 3.2 mm). To ensure proper placement, each wire bundle was encased in a 27-gauge stainless-steel tubing and trimmed using fine spring scissors. The trimmed wires extended approximately 1.5–2 mm beyond the tubing's end. Initially, the wire tips were positioned 4.2 mm ventral from the brain surface. Following the surgical procedure, the rats received oral doses of Cephalexin (15 mg/kg) twice daily for a duration of 2 weeks to prevent any potential infections.

## Catheter surgery

Rats used for cocaine self-administration received chronic indwelling jugular catheter implants (Instech Laboratories). Rats were anesthetized using ketamine (100 mg/kg, i.p., Sigma) and xylazine (10 mg/kg, i.p., Sigma). Blunt dissection was performed to isolate right external jugular veins, and catheters were surgically implanted 3 cm into the veins. Catheters were passed subcutaneously to the back, where they were attached to an external harness. Carprofen (5 mg/kg, s.c., Pfizer) was administered after surgery as an analgesic. Rats recovered for 7 days before self-administration began. During recovery and self-administration, catheters were flushed daily with a cocktail of enrofloxacin (4.0 mg/ml, Bayer) and heparinized saline (50 IU/ml in 0.9% sterile saline, Sigma) to maintain catheter patency.

## Self-administration

Following recovery from catheterization surgery, rats were trained to self-administer cocaine-HCl (0.75 mg/kg/infusion; $n = 3$) or sucrose (10% wt/vol; $n = 3$) for 14 consecutive days. Rats were trained in modular behavioral test chambers (Coulbourn Instruments) housed in sound-attenuating boxes. Each chamber was equipped with two levers positioned 8 cm above the floor on opposite sides of the same wall. For intravenous cocaine self-administration, catheter ports were attached to silastic tubing connected to infusion pumps (Med Associates Inc) located outside sound-attenuating boxes. For sucrose self-administration, sucrose solution was delivered via photobeam-monitored recessed dippers. Daily sessions were 3 hr and began with the illumination of a house light and the insertion of an active lever. Under a fixed ratio 1 (FR1) schedule of reinforcement, active lever presses resulted in 4-s infusions or dipper insertions (0.05 ml), for cocaine-HCl or sucrose, respectively, and were paired with the illumination of a cue light above the active lever. Infusions and dipper insertions were followed by a 40-s timeout period when the active lever retracted and the house light was extinguished. Following the timeout period, the lever was reinserted and the house light was turned back on. Inactive lever presses had no programmed consequence. Reinforcers were limited to 20 per hour to prevent overdose in cocaine self-administering rats. When 20 reinforcers were earned in less than an hour, a timeout period as described above was imposed until the beginning of the next hour.

## Single-unit recording

Electrophysiological signals were recorded using Plexon OmniPlex systems (Plexon, Dallas, TX). These signals were digitized, amplified, and subjected to bandpass filtering (250–8000 Hz) to isolate spike activity. Manual thresholding was performed on each active channel to capture unsorted spikes. Timestamps for behavioral events were synchronized with the Plexon system and recorded together with the neural activity. To remove noise and identify single units, spike sorting was carried out offline using Offline Sorter (v.4.0; Plexon), utilizing a template-matching algorithm. The sorted files were then processed in NeuroExplorer (Nex Technologies, Colorado Springs, CO) to extract timestamps for both unit activity and behavioral events. Subsequently, these timestamps were exported as MATLAB (2021b; MathWorks) formatted files for further analysis. It is important to note that the electrodes were not advanced within a specific problem. However, we cannot make any claims regarding the consistency of single units recorded on different days within the same problem, as they may represent distinct neurons. To sample different neural populations during odor problems, the electrodes were advanced by approximately 120 μm. Both in vivo recordings and spike sorting were performed in a blinded manner, without knowledge of whether the subject belonged to the Sucrose or Cocaine group.

## Quantification and statistical analyses

Quantification and statistical analyses were conducted using MATLAB (R2024b; MathWorks) and Python Software Foundation, 2024. The sample sizes of rats and neurons were not predetermined through specific statistical methods; however, they are consistent with those reported in previous studies conducted by our lab and other research groups.

### Task events and peri-event spike train analysis

The trials were divided into nine distinct epochs, each corresponding to different task events: 'ITI-a', 'Light', 'Poke', 'Odor', 'Unpoke', 'Choice', 'Outcome', 'postOut', and 'ITI-b'. 'ITI-a' represented the

time point 0.7 s before the house-light turned on. On reward trials, the well-entry moment was labeled 'Choice'. The 'Outcome' epoch denoted the time of reward delivery. On non-reward trials, the end of the 2-s response window was marked as 'Choice', and a time point 0.7 s after 'Choice' was labeled as 'Outcome'. Both on reward and non-reward trials, 0.7 s after the outcome was recorded as 'postOut', followed by another 0.7 s designated as 'ITI-b'. Behavioral performance was evaluated by calculating the percentage of trials in which the rats responded correctly and determining the latency at which they initiated a trial after the onset of the light. The spike train for each isolated single unit was aligned to the onset of each task event to create a peri-event time histogram (PETH). The PETH was constructed with a pre-event time of 200 ms and a post-event time of 600 ms, counting the number of spikes within each 100 ms bin. To smooth the PETH on each trial, a Gaussian kernel with a $\sigma$ (standard deviation) of 50 ms was applied. For further analysis, a random selection of 30 correct trials was made from each trial type, resulting in a total of 240 trials (30 trials × 8 trial types). The post-event firing rates (100–600 ms) were averaged to obtain a single measure of neural activity for each neuron on each trial during each task epoch.

## Classification analyses

The neural data collected during each task epoch were organized into a two-dimensional matrix, where the rows represented individual trials and the columns represented the firing rates of each neuron across all trials. In other words, each trial was represented as a vector, with each dimension corresponding to the firing rate of a specific neuron. Neurons recorded across different sessions were concatenated, aligning them with the corresponding trials to create pseudoensembles. To remove temporal correlations between neurons and generate different pseudoensembles, we shuffled the trial orders within each trial type. This shuffling process was repeated 10,000 times, resulting in 10,000 pseudoensembles. By using the linear Support Vector Machine (SVM) for classification analyses, we assessed the classification accuracy through a leave-one-out cross-validation procedure. Specifically, one trial from each trial type was excluded for future testing, while the remaining trials were used to train the classifier. For each pseudoensemble, the leave-one-out cross-validation was repeated 200 times to estimate the mean decoding accuracy. The decoding analyses were conducted on the 10,000 pseudoensembles to calculate an overall mean decoding accuracy. To determine the statistical significance of the overall mean decoding accuracy, we estimated a 95% confidence interval by running the same decoding process with label-shuffled pseudoensembles.

## Cross-sequence decoding

To assess population decoding of position within and across sequences in OFC cells ($n$ = 1000), we applied an SVM classifier. Decoding accuracy was estimated using a leave-one-out cross-validation approach. In each iteration, 30 trials per trial type were randomly sampled for each of the 9 task epochs, producing a 120 (trials) × 9 (epochs) matrix per sequence. One trial from each trial type in sequence 1 was withheld for testing, while the trial with the corresponding index in sequence 2 was simultaneously set aside for across-sequence evaluation. The classifier was trained on the remaining sequence 1 trials. For each epoch and trial type, decoding accuracy was averaged across 1000 iterations to obtain the mean performance.

## TCA analysis

To perform TCA, neuronal firing rates for each group were structured into a three-dimensional array ($N \times T \times K$), where $N$ represents the number of neurons, $T$ the time samples *per* trial, and $K$ the number of experimental trials. This array, referred to as a third-order tensor, captures neuronal activity across trials. Data were exported as a MATLAB .mat file and imported into Spyder for analysis using the TensorTools package (*Williams et al., 2018*). Temporal factors were grouped as follows: factors 1–8 (preTrial), 9–16 (Light), 17–24 (Poke), 25–32 (Odor), 33–40 (Unpoke), 41–48 (Choice), 49–56 (Outcome), 57–64 (postTrial1), and 65–72 (postTrial2). There are 8 trial types, each with 40 trials. Trial factors 1–80 and 241–320 represent positive trials with reward, while 81–240 correspond to negative trials without reward. Trials 1–80 map to P1, 81–160 to P2, 161–240 to P3, and 241–320 to P4. Error and similarity plots were generated to assess the stability of TCA's optimization landscape. Specifically, we ran the TCA optimization algorithm 100 times for each of 10 components, initializing each run with random conditions, and plotted the normalized reconstruction error across all iterations. This

approach allowed us to evaluate whether certain runs converged to local minima with high reconstruction errors. Additionally, similarity scores were computed for each model relative to the best-fit model with the same number of components, with the lines representing the mean similarity as a function of component count. Across all tested component numbers, the 100 repeated runs exhibited substantial overlap and consistently produced similarity scores above 0.8, demonstrating high quantitative consistency. MI was calculated for across-trial factors and trial type, as well as across-trial factors and time, after aligning components between the Sucrose and Cocaine groups. The alignment followed the same ordering of across-temporal factors between the two groups. MI was calculated for across-trial factors and trial type, as well as across-trial factors and time, after aligning components between the Sucrose and Cocaine groups. The alignment followed the same ordering of across-temporal factors between the two groups.

## Acknowledgements

This research was funded by the Intramural Research Program at the National Institute on Drug Abuse (ZIA-DA000587). The views expressed in this article are solely those of the authors and do not necessarily represent the opinions of the NIH or DHHS.

## Additional information

### Funding

| Funder | Grant reference number | Author |
|---|---|---|
| National Institute on Drug Abuse | Z1A-DA000587 | Geoffrey Schoenbaum |

The funders had no role in study design, data collection, and interpretation, or the decision to submit the work for publication.

### Author contributions

Wenhui Zong, Conceptualization, Data curation, Formal analysis, Investigation, Writing – original draft, Writing – review and editing; Lauren Mueller, Data curation, Investigation; Zhewei Zhang, Investigation, Methodology; Jinfeng Zhou, Conceptualization, Data curation, Formal analysis, Investigation; Geoffrey Schoenbaum, Conceptualization, Resources, Supervision, Investigation, Methodology, Writing – original draft, Project administration, Writing – review and editing

### Author ORCIDs

Wenhui Zong ⓘ https://orcid.org/0000-0003-3792-6029
Lauren Mueller ⓘ https://orcid.org/0000-0001-6179-2759
Zhewei Zhang ⓘ https://orcid.org/0000-0001-5841-4239
Jinfeng Zhou ⓘ https://orcid.org/0000-0003-1893-1025
Geoffrey Schoenbaum ⓘ https://orcid.org/0000-0001-8180-0701

### Ethics

All behavioral testing was conducted at the NIDA-IRP, and the animal care and experimental procedures were conducted in accordance with the guidelines set by the US National Institutes of Health (NIH) and approved by the Animal Care and Use Committee (ACUC) at the NIDA-IRP.

Reviewer #1 (Public review): https://doi.org/10.7554/eLife.109883.3.sa1
Reviewer #2 (Public review): https://doi.org/10.7554/eLife.109883.3.sa2
Author response https://doi.org/10.7554/eLife.109883.3.sa3

## Additional files

### Supplementary files

MDAR checklist

## Data availability

Data and code availability: All data and analysis code associated with this study are available on OSF at https://osf.io/azvhm/.

The following dataset was generated:

| Author(s) | Year | Dataset title | Dataset URL | Database and Identifier |
|---|---|---|---|---|
| Zong W | 2026 | Prior cocaine use disrupts identification of hidden states by single units and neural ensembles in orbitofrontal cortex | https://osf.io/azvhm/ | Open Science Framework, azvhm |

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
