## [Editor Report · eLife Assessment]

This **fundamental** work shows that a history of cocaine self-administration disrupts the orbitofrontal cortex's ability to encode similarities between distinct sensory stimuli that possess identical task information—hidden states. The evidence supporting these conclusions is **compelling**, with methods and analyses spanning self-administration, a novel 'figure 8' sequential odor task, recordings from 3,881 single units, and sophisticated firing analyses revealing complex orbitofrontal representations of task structure. These results will be of broad interest to psychologists, neuroscientists, and clinicians.

---

## [Referee Report · Reviewer #1 (Public review)]

Summary:

In this study, the authors trained rats on a "figure 8" go/no-go odor discrimination task. Six odor cues (3 rewarded and 3 non-rewarded) were presented in a fixed temporal order and arranged into two alternating sequences that partially overlap (Sequence #1: 5^+^-0^-^-1^-^-2^+^; Sequence #2: 3^+^-0^-^-1^-^-4^+^) --forming an abstract figure-8 structure of looping odor cues.

This task is particularly well-suited for probing representations of hidden states, defined here as the animal's position within the task structure beyond superficial sensory features. Although the task can be solved without explicit sequence tracking, it affords the opportunity to generalize across functionally equivalent trials (or "positions") in different sequences, allowing the authors to examine how OFC representations collapse across latent task structure.

Rats were first trained to criterion on the task and then underwent 15 days of self-administration of either intravenous cocaine (3 h/day) or sucrose. Following self-administration, electrodes were implanted in lateral OFC, and single-unit activity was recorded while rats performed the figure-8 task.

Across a series of complementary analyses, the authors report several notable findings. In control animals, lOFC neurons exhibit representational compression across corresponding positions in the two sequences. This compression is observed not only in trial/positions involving overlapping odor (e.g., Position 3 = odor 1 in sequence 1 vs sequence 2), but also in trials/positions involving distinct, sequence-specific odors (e.g., Position 4: odor 2 vs odor 4) --indicating generalization across functionally equivalent task states. Ensemble decoding confirms that sequence identity is weakly decodable at these positions, consistent with the idea that OFC representations collapse incidental differences in sensory information into a common latent or hidden state representation. In contrast, cocaine-experienced rats show persistently stronger differentiation between sequences, including at overlapping odor positions.

Strengths:

- Elegant behavioral design that affords the detection of hidden-state representations.

- Sophisticated and complementary analytical approaches (single-unit activity, population decoding, and tensor component analysis).

Weaknesses:

-The number of subjects is small --can't fully rule out idiosyncratic, animal-specific effects.

Comments on revisions:

The authors have thoroughly addressed all of my previous comments. Congratulations on an excellent paper!

---

## [Referee Report · Reviewer #2 (Public review)]

In the current study, the authors use an odor-guided sequence learning task described as a "figure 8" task to probe neuronal differences in latent state encoding within the orbitofrontal cortex after cocaine (n = 3) vs sucrose (n = 3) self-administration. The task uses six unique odors which are divided into two sequences that run in series. For both sequences, the 2nd and 3rd odors are the same and predict reward is not available at the reward port. The 1st and 4th odors are unique, and are followed by reward. Animals are well-trained before undergoing electrode implant and catheterization, and then retrained for two weeks prior to recording. The hypothesis under test is that cocaine-experienced animals will be less able to use the latent task structure to perform the task, and instead encode information about each unique sequence that is largely irrelevant. Behaviorally, both cocaine and sucrose-experienced rats show high levels of accuracy on task, with some group differences noted. When comparing reaction times and poke latencies between sequences, more variability was observed in the cocaine-treated group, implying animals treated these sequences somewhat differently. Analyses done at the single unit and ensemble level suggests that cocaine self-administration had increased the encoding of sequence-specific information, but decreased generalization across sequences. For example, the ability to decode odor position and sequence from neuronal firing in cocaine-treated animals was greater than controls. This pattern resembles that observed within the OFC of animals that had fewer training sessions. The authors then conducted tensor component analysis (TCA) to enable a more "hypothesis agnostic" evaluation of their data.

Overall, the paper is well written and the authors do a good job of explaining quite complicated analyses so that the reader can follow their reasoning. The findings are important, and the results are compelling. The introduction and discussion contextualize the experiments in the context of the literature, and explain the novelty and significance of the current findings. Specifically, the observation that cocaine self-administration impairs generalization across task sequences at the single unit level builds on previous observations of aberrant neuronal activity within the OFC in animals with a history of cocaine self-administration. These new data point to a neurophysiological mechanism that could explain why drug-seeking is so context dependent, and hard to ameliorate with therapeutic strategies that take place within a clinical setting.

The authors clearly acknowledge the major limitations of this work, namely that the sample size is restricted due to the technical challenges of performing in vivo electrophysiology recordings combined with self-administration, and that animals of only one sex were used. Importantly, the data from all rats within each group was remarkably homogeneous, increasing confidence in the conclusions drawn.

---

## [Author Response]

The following is the authors’ response to the original reviews

**Public Reviews:**

**Reviewer #1 (Public review):**
Summary:In this study, the authors trained rats on a "figure 8" go/no-go odor discrimination task. Six odor cues (3 rewarded and 3 non-rewarded) were presented in a fixed temporal order and arranged into two alternating sequences that partially overlap (Sequence #1: 5^+^-0^-^-1^-^-2^+^; Sequence #2: 3^+^-0^-^-1^-^-4^+^) - forming an abstract figure-8 structure of looping odor cues.This task is particularly well-suited for probing representations of hidden states, defined here as the animal's position within the task structure beyond superficial sensory features. Although the task can be solved without explicit sequence tracking, it affords the opportunity to generalize across functionally equivalent trials (or "positions") in different sequences, allowing the authors to examine how OFC representations collapse across latent task structure.Rats were first trained to criterion on the task and then underwent 15 days of self-administration of either intravenous cocaine (3 h/day) or sucrose. Following self-administration, electrodes were implanted in lateral OFC, and single-unit activity was recorded while rats performed the figure-8 task.Across a series of complementary analyses, the authors report several notable findings. In control animals, lOFC neurons exhibit representational compression across corresponding positions in the two sequences. This compression is observed not only in trial/positions involving overlapping odor (e.g., Position 3 = odor 1 in sequence 1 vs sequence 2), but also in trials/positions involving distinct, sequence-specific odors (e.g., Position 4: odor 2 vs odor 4) - indicating generalization across functionally equivalent task states. Ensemble decoding confirms that sequence identity is weakly decodable at these positions, consistent with the idea that OFC representations collapse incidental differences in sensory information into a common latent or hidden state representation. In contrast, cocaine-experienced rats show persistently stronger differentiation between sequences, including at overlapping odor positions.Strengths:Elegant behavioral design that affords the detection of hidden-state representations.Sophisticated and complementary analytical approaches (single-unit activity, population decoding, and tensor component analysis).Weaknesses:The number of subjects is small - can't fully rule out idiosyncratic, animal-specific effects.Comments(1) Emergence of sequence-dependent OFC representations across learning.A conceptual point that would benefit from further discussion concerns the emergence of sequence-dependent OFC activity at overlapping positions (e.g., position P3, odor 1). This implies knowledge of the broader task structure. Such representations are presumably absent early in learning, before rats have learned the sequence structure. While recordings were conducted only after rats were well trained, it would be informative if the authors could comment on how they envision these representations developing over learning. For example, does sequence differentiation initially emerge as animals learn the overall task structure, followed by progressive compression once animals learn that certain states are functionally equivalent? Clarifying this learning-stage interpretation would strengthen the theoretical framing of the results.

We agree that the emergence of sequence-dependent OFC activity at overlapping positions (e.g., P3) implies knowledge of the broader task structure and therefore must depend on learning. Although we did not record during early acquisition in the current study, we can outline a learning-stage framework consistent with both prior work and the comparative analyses included here and include it in the discussion.

We think the development of OFC representations is a multi-stage process. Early in learning, before animals have acquired the sequential structure of the task, OFC activity is likely dominated by local sensory features and immediate reinforcement history, with little differentiation between sequences at overlapping positions. As animals learn that odors are embedded within extended sequences that have utility for predicting future outcomes, OFC representations would begin to differentiate identical sensory cues based on their sequence context, giving rise to sequence-dependent activity at positions such as P3. This stage reflects acquisition of the broader task structure and the recognition that current cues carry information about future states.

With continued training, however, OFC representations normally undergo a further refinement: positions that differ in sensory identity but are functionally equivalent become compressed, while distinctions that are irrelevant for guiding behavior are suppressed. Evidence for this later stage comes from our over-trained control animals, in which discrimination between overlapping positions is near chance across most trial epochs, and from prior work using the same task in less-trained animals, where sequence-dependent discrimination is more strongly preserved. Thus, sequence differentiation appears to emerge during structure learning but is subsequently down weighted as animals learn which distinctions are behaviorally irrelevant.

Within this framework, prior cocaine exposure appears to interfere specifically with this later refinement stage. Cocaine-experienced rats exhibit OFC representations resembling those seen earlier in learning—retaining sequence-dependent discrimination at overlapping and functionally equivalent positions—despite extensive training. This suggests not a failure to acquire task structure per se, but rather an impairment in the ability to collapse across states that share common underlying causes.

(2) Reference to the 24-odor position taskThe reference to the previously published 24-odor position task is not well integrated into the current manuscript. Given that this task has already been published and is not central to the main analyses presented here, the authors may wish to (a) better motivate its relevance to the current study or (b) consider removing this supplemental figure entirely to maintain focus.

Thanks for your suggestion, we have removed this supplemental figure as suggested.

(3) Missing behavioral comparisonLine 117: the authors state that absolute differences between sequences differ between cocaine and sucrose groups across all three behavioral measures. However, Figure 1 includes only two corresponding comparisons (Fig. 1I-J). Please add the third measure (% correct) to Figure 1, and arrange these panels in an order consistent with Figure 1F-H (% correct, reaction time, poke latency).

Thanks for your suggestion, we have included the related figure as suggested.

(4) Description of the TCA componentLine 220: authors wrote that the first TCA component exhibits low amplitude at positions P1 and P4 and high amplitude at positions P2 and P3. However, Figure 3 appears to show the opposite pattern (higher magnitude at P1 and P4 and lower magnitude at P2 and P3). Please check and clarify this apparent discrepancy. Alternatively, a clearer explanation of how to interpret the temporal dynamics and scaling of this component in the figure would help readers correctly understand the result.

Thanks for your suggestion. We appreciate this point and agree that clearer guidance on how to interpret the temporal and scaling properties of the tensor components would help readers. In the TCA framework, each component is defined by three separable factors: a neuron factor, a temporal factor, and a trial (position) factor. The temporal factor reflects the shape of the activity pattern within a trial, indicating when during the trial that component is expressed, whereas the trial factor reflects how strongly that temporal pattern is expressed at each position and across trials.

Importantly, the absolute scaling of these factors is not independently meaningful. Because TCA components are scale-indeterminate, the magnitude of the temporal factor and the trial factor should be interpreted relative to one another within a component, not across components. Thus, a large value in the trial factor does not imply stronger neural activity per se, but rather greater expression of that component’s characteristic temporal pattern at that position or trial.

Accordingly, when a component shows similar temporal dynamics across groups but differs in its trial factor structure—as observed here—the interpretation is that the same within-trial dynamics are being differentially recruited across task positions, rather than that the timing of neural responses has changed.

We have added a brief discussion of this in this section of the results in the manuscript.

(5) Sucrose controlSucrose self-administration is a reasonable control for instrumental experience and reward exposure, but it means that this group also acquired an additional task involving the same reinforcer. This experience may itself influence OFC representations and could contribute to the generalization observed in control animals. A brief discussion of this possibility would help contextualize the interpretation of cocaine-related effects.

We agree that sucrose self-administration is not a perfect neutral manipulation and that this experience could, in principle, influence OFC representations. In particular, sucrose self-administration involves instrumental responding for the same primary reinforcer used in the odor task, and thus may promote additional learning about reward predictability, action–outcome contingencies, or contextual structure that could facilitate generalization.

Several considerations, however, suggest that the generalization observed in control animals primarily reflects learning-dependent refinement of task representations rather than a specific consequence of sucrose self-administration per se. First, the amount of sucrose administered during this phase was minimal (50 µl × 60 presses at most per session for 14 sessions) compared with the total sucrose reward obtained during task recording (100 µl × 160 trials per session for several dozen sessions). Second, all rats were extensively trained on the odor sequence task prior to any self-administration, and the key signatures of compression and generalization we report—near-chance discrimination between functionally equivalent positions—are consistent with prior studies using the same task in animals that did not undergo sucrose self-administration. Finally, comparisons to less-trained animals in earlier work show that OFC representations evolve toward greater abstraction with increasing task experience, indicating that generalization is a property of advanced learning rather than a unique outcome of sucrose exposure.

Importantly, even if sucrose self-administration were to enhance generalization in OFC, this would not account for the primary finding that cocaine-experienced rats fail to show these signatures despite identical task training and parallel instrumental experience. Thus, the critical comparison is not between sucrose-trained animals and naive controls, but between two groups matched for self-administration experience, differing only in the pharmacological consequences of the reinforcer. Within this framework, the absence of position-general representations in cocaine-experienced rats reflects a disruption of normal learning-dependent abstraction rather than an artifact of the control condition.

We have added a brief discussion acknowledging that sucrose self-administration may bias OFC toward abstraction, while emphasizing that cocaine exposure prevents the emergence or maintenance of these representations under otherwise comparable experiential conditions.

(6) Acknowledge low NThe number of rats per group is relatively low. Although the effects appear consistent across animals within each group, this sample size does not fully rule out idiosyncratic, animal-specific effects. This limitation should be explicitly acknowledged in the manuscript.

We acknowledge that the number of animals per group is relatively small and therefore cannot fully rule out animal-specific effects. However, the key neural and behavioral signatures reported here were consistent across individual animals within each group and across multiple levels of analysis, and no outliers were observed. In addition, sample sizes of this scale are common in cocaine self-administration studies due to their technical and logistical constraints. We did not attempt to obscure this limitation and have now explicitly acknowledged it in the manuscript discussion.

(7) Figure 3E-F: The task positions here are ordered differently (P1, P4, P2, P3) than elsewhere in the paper. Please reorder them to match the rest of the paper.

Thank you for pointing this out. We agree that the ordering of task positions in Figures 3E–F should be consistent with the rest of the manuscript. We have reordered the positions to match the standard sequence order used elsewhere in the paper (P1, P2, P3, P4) to improve clarity and avoid confusion.

**Reviewer #2 (Public review):**
In the current study, the authors use an odor-guided sequence learning task described as a "figure 8" task to probe neuronal differences in latent state encoding within the orbitofrontal cortex after cocaine (n = 3) vs sucrose (n = 3) self-administration. The task uses six unique odors which are divided into two sequences that run in series. For both sequences, the 2nd and 3rd odors are the same and predict reward is not available at the reward port. The 1st and 4th odors are unique, and are followed by reward. Animals are well-trained before undergoing electrode implant and catheterization, and then retrained for two weeks prior to recording. The hypothesis under test is that cocaine-experienced animals will be less able to use the latent task structure to perform the task, and instead encode information about each unique sequence that is largely irrelevant. Behaviorally, both cocaine and sucrose-experienced rats show high levels of accuracy on task, with some group differences noted. When comparing reaction times and poke latencies between sequences, more variability was observed in the cocaine-treated group, implying animals treated these sequences somewhat differently. Analyses done at the single unit and ensemble level suggests that cocaine self-administration had increased the encoding of sequence-specific information, but decreased generalization across sequences. For example, the ability to decode odor position and sequence from neuronal firing in cocaine-treated animals was greater than controls. This pattern resembles that observed within the OFC of animals that had fewer training sessions. The authors then conducted tensor component analysis (TCA) to enable a more "hypothesis agnostic" evaluation of their data.Overall, the paper is well written and the authors do a good job of explaining quite complicated analyses so that the reader can follow their reasoning. I have the following comments.While well-written, the introduction mainly summarises the experimental design and results, rather than providing a summary of relevant literature that informed the experimental design. More details regarding the published effects of cocaine self-administration on OFC firing, and on tests of behavioral flexibility across species, would ground the paper more thoroughly in the literature and explain the need for the current experiment.

We appreciate this suggestion and have tried to expand the Introduction to more explicitly situate the study within the existing literature on cocaine-induced changes in OFC function. In particular, prior work has shown that cocaine self-administration alters OFC firing properties and disrupts behavioral flexibility across species, including impairments in reversal learning, outcome devaluation, and sensory preconditioning. We have revised the Introduction to expand this literature review and more clearly articulate how these established findings motivated our focus on OFC representations of hidden task structure and generalization.

For Fig 1F, it is hard to see the magnitude of the group difference with the graph showing 0-100%- can the y axis be adjusted to make this difference more obvious? It looks like the cocaine-treated animals were more accurate at P3- is that right?The concluding section is quite brief. The authors suggest that the failure to generalize across sequences observed in the current study could explain why people who are addicted to cocaine do not use information learned e.g. in classrooms or treatment programs to curtail their drug use. They do not acknowledge the limitations of their study e.g. use of male rats exclusively, or discuss alternative explanations of their data.

We agree that the current 0–100% scale can make small differences difficult to discern. We will make it clear in the figure captions (We will adjust the y-axis to a narrower range to better highlight group differences). Across P3, cocaine-experienced rats were more accurate than controls.

We appreciate the suggestion to expand the discussion. We have revised the concluding section to acknowledge key limitations, including the use of only male rats, the number of subjects, and to note that alternative explanations—such as differences in motivational state or attention—could also contribute to the observed effects. These revisions provide a more balanced interpretation while retaining the focus on OFC-mediated generalization as a potential mechanism for persistent, context-specific drug-seeking.

Is it a problem that neuronal encoding of the "positions" i.e. the specific odors was at or near chance throughout in controls? Could they be using a simpler strategy based on the fact that two successive trials are rewarded, then two successive trials are not rewarded, such that the odors are irrelevant?

We thank the reviewer for this point. While neuronal encoding of individual positions (specific odors) in control animals was comparatively lower, this does not indicate that the rats were using a simpler strategy based solely on reward patterns. First, rats were extensively trained on the odor sequence task prior to recordings, demonstrating accurate discrimination across all positions, and their trial-by-trial behavior reflects sensitivity to specific odors rather than only reward alternation. Second, the task design—with overlapping sequences and positions that differ in reward contingency across sequences—requires tracking odor-specific context to maximize reward; a purely “two rewarded, two non-rewarded” strategy would fail at overlapping positions and would not account for the compression of functionally equivalent positions observed in the OFC. Third, in the less-trained rats shown in Figure 3C, decoding accuracy was higher than in the sucrose group, indicating that these animals still differentiated negative positions. With additional training, decoding patterns suggested improved generalization across positions. Thus, the near-chance neural selectivity in controls reflects representation of latent task states rather than external sensory cues, consistent with the idea that OFC abstracts task-relevant structure and ignores irrelevant sensory differences.

When looking at the RT and poke latency graphs, it seems the cocaine-experienced rats were faster to respond to rewarded odors, and also faster to poke after P3. Does this mean they were more motivated by the reward?

At present, the basis of these response-time differences remains unclear, in part because motivation is difficult to define operationally. If motivation is indexed solely by reaction time or poke latency, then the data are consistent with increased response vigor in cocaine-experienced rats. Indeed, RT and poke-latency measures indicate that cocaine-experienced rats responded more quickly on some rewarded trials, including after P3. However, overall task performance was high in both groups, suggesting that these differences cannot be attributed simply to superior learning or engagement. Faster responses may also reflect differences in deliberation or strategy, with cocaine-experienced rats relying more on rapid, stimulus-driven responding and sucrose-trained rats engaging in more careful evaluation. In addition, altered reward sensitivity or persistent effects of cocaine exposure may contribute to these behavioral differences. Thus, the faster responses observed in cocaine-experienced rats likely reflect a combination of heightened reward responsivity and altered encoding of task structure, rather than a straightforward increase in motivation alone.

**Recommendations for the authors:**
The reviewers were very positive about the manuscript and emphasized the rigor and state of the art analyses. Two points that came up were the very small n (6 total and 3 per condition) and the exclusive use of males. Adding more subjects is not recommended. However, more discussion and acknowledgement of this issue is recommended. The main concern is that idiosyncratic differences between individuals (not differences in cocaine history) are responsible for the differences observed in OFC encoding.

We acknowledge that the sample size (n = 3 per group) and use of only male rats limit generalizability and do not fully rule out idiosyncratic, individual-specific effects. However, the key neural and behavioral signatures we report were consistent across all animals within each group and across multiple analyses (single-unit, ensemble decoding, and TCA). We now explicitly note these limitations in the Discussion, emphasizing that while individual variability cannot be fully excluded, the convergence of results across multiple levels of analysis supports the interpretation that the observed differences reflect effects of prior cocaine exposure rather than idiosyncratic differences.

**Reviewer #2 (Recommendations for the authors):**
In the legend to figure 2, the authors state "Notably, rats could discriminate between the two sequences (S1 vs. S2) based solely on current sensory information at two task epochs ["Odor" at P3 and P4; black bars]. At all other task epochs, indicated by gray bars, the discrimination relied on an internal memory of events". I'm confused by this statement- how does the odor at P3 help to discriminate the sequences? Surely P1 and P4 are the times when the odor sampling indicates which sequence they are in?

We thank the reviewer for pointing out this source of confusion. The statement in the original figure legend was imprecise, and we have removed the figure and revised the figure legends because the results in the left panel substantially overlapped with those shown in the right panel. In this task, odors at positions P1 and P4 are the only cues that directly signal sequence identity, whereas the odors presented at P2 and P3 are identical across sequences. Accordingly, discrimination observed during the “Odor” epoch at P3 does not reflect sensory differences but instead depends on the animal’s use of internal memory or sequence context to infer sequence identity.